# Understanding and acceptance of the theory of evolution in high school students in Mexico

**Guadalupe Salazar-Enriquez** [ID]**, Jose Rafael Guzman-Sepulveda** [ID]**, Gonzalo Peñaloza** [ID] *

Centro de Investigación y de Estudios Avanzados del IPN, Unidad Monterrey, Apodaca, Nuevo León, México

* g.pjimenez@cinvestav.mx

## Abstract

The Theory of Evolution (TE) is the backbone of biology and is the best way to explain the diversity of species that exist on the planet. However, despite all the supporting evidence, TE remains poorly understood and accepted. In this study, the levels of acceptance and understanding of TE were measured, respectively, using the Inventory of Student Evolution Acceptance (I-SEA) and Knowledge of Evolution Exam (KEE) questionnaires, in high school students in Monterrey, Mexico (N = 370). The results show that the acceptance of TE ranges from moderate (90.3 out of 120) to high (3.7 out of 5), depending on the scale with which it is measured, while the level of comprehension is low (4.5 out of 10). Statistical analysis of the data collected reveals that there is a positive relationship between acceptance and understanding of TE (r = 0.34). In addition, the proportions of I-SEA and KEE were evaluated based on several factors, such as religion and educational level of the parents, among others. It was found that the level of education of the parents positively affects the understanding of the basic concepts of TE, while religion is the main factor of negative influence on both acceptance and understanding. Finally, the low comprehension shown in this study suggests a revision and readjustment of the contents that are taught in the upper secondary education curriculum.

**Data Availability Statement:** All relevant data are within the article and its Supporting Information files.

## 1. Introduction

The Theory of Evolution (TE) is a fundamental aspect of modern science [1]. It is essential to explain biological diversity on the planet, as well as to the classification and identification of species [2]. However, despite constituting a central pillar of biology and being part of the school curriculum in many countries [3], the TE is poorly understood and has relatively low acceptance [1, 4–6].

Many studies have reported on the difficulties of teaching and learning the TE, including misconceptions of teachers [5, 7–9]; the short time devoted to these topics during a school year [10–14]; and, the influence of religious beliefs [6, 8, 15–17], among others. It has been shown that these limitations strongly play a major role in both accepting and understanding the basic concepts of the TE. In fact, besides trying to overcome the above-mentioned limitations, educational researchers have dedicated themselves to unveil the connection between *understanding* and *acceptance*, as well as find ways where one can help the other.

**Funding:** G.S This work was supported by CONACYT (Consejo Nacional de Ciencia y Tecnología) https://conacyt.mx/becas_posgrados/ The funders had no role in study design, data collection and analysis, decision to publish, or preparation of the manuscript.

**Competing interests:** The authors have declared that no competing interests exist.

## 1.1 Acceptance of evolution

In the context of communication and education of biological evolution, it is important to distinguish between two key concepts: *belief* and *acceptance* [18]. Acceptance is based on the validity of a construct, its plausibility to explain a phenomenon, and it is based on evidence; on the other hand, a belief is based on personal convictions, faith, feelings, and opinions [19]. Thus, since the TE is a scientific theory, the main goal should not be for people to believe in it, but to evaluate, test, and accept it as the best known explanation that is supported by solid evidence [20].

In this sense, science educators have developed different tools to measure the acceptance of the TE. A well-known tool is the Measure of Acceptance of the Theory of Evolution (MATE), which is a 20-question questionnaire, developed by Rutledge and Warden [21] based on a Likert scale, that has been widely used in research [18]. Deeper analysis has shown that acceptance of the TE depends on the sublevel of evolution: macroevolution, microevolution, and human evolution [18]. Macroevolution is understood as the result of long-term processes that originate new species, where natural selection, among other mechanisms, plays a key role, while microevolution refers to the change in allele frequencies within a population [22]. It has been observed, for instance, that some people accept well the notion of internal variations within a species, but do not accept the origin of new ones.

Research on the acceptance of the TE has focused on students [1, 11, 13, 14, 23–26], teachers [5, 7–9, 12, 27], and the public [4, 6]. Overall, moderate acceptance has been reported [6], but levels vary between groups and due to many factors. For instance, it has been found to be strongly linked to religious beliefs [28], especially in religions with a creation story, particularly Muslim, Christian, and Abrahamic [6]. Study [24] carried out research in the UK with first-year science majors and found that students who identified themselves with a certain religion had lower levels of acceptance than those who did not identified with any religion. This inverse relationship between acceptance and religion has been widely documented [4, 6, 17, 24, 28–30]. Those for which religion represents an important part of their lives are more likely to reject the TE, despite acknowledging that it is scientifically valid [17].

Sociocultural context is also relevant when dealing with the acceptance of the TE because its accounts for factors that can shape the attitudes of students towards science [31]. It has been observed, for instance, that positive attitudes towards science are correlated with a higher degree of acceptance of the TE [1]. Conversely, the accumulation of negative experiences in science, either due to a conflict of ideas [32], conceptual difficulties [33], or pedagogical and affective factors [34], can lead the student to have a low sense of belonging in scientific areas, making it difficult to retain their participation in STEM areas [Science, Technology, Engineering and Mathematics) [35]. In this sense, in a study with university students, Rice et al. [23] found a well-defined relationship between the field of study and the acceptance of the TE, showing that students from STEM programs, specially biology-related, had higher levels of acceptance. Interestingly, a positive relationship between the acceptance of the TE of students and the educational level of their parents has also been observed (higher academic level of the parents was associated to higher acceptance of the students), indicating that positive attitudes towards science are also largely determined by the educational background of the parents [1, 5]

## 1.2 Understanding of evolution

One of the main goals of science education is fostering the understanding of scientific processes and concepts by students [20]. However, defining scientific knowledge is a cornerstone to specify the limits and ways to teach science; it is not a simple task. Broncano [36] defines

knowledge like a true belief into an epistemological agent. He argues that such belief should be plausible for the subject, using his own plausibility criteria. On the other hand, Smith and Siegel [20], agree that knowledge is dynamic and must be evaluated through evidence like methodological criteria that every could be shared by any person.

Cobern [37] claims that it is difficult to leave the belief outside the classroom, simply arguing that "it does not support the evidence", since what is constituted as evidence for each subject is relative and is subordinated to its initial assumptions. He proposes discussing in the classroom the meaning of scientific evidence and the assumptions on which science is founded, in order to demarcate between (scientific) knowledge and belief.

Students usually build their own understanding of the any topic [38]. According to some authors, the understanding of the subject occurs when knowledge is transformed, when subjects go from thinking about *what happens* to thinking about *how it happens*. For example, we could know what eclipses are, but it is until we understand *how* they occur, when we understand the phenomenon [39]. In this way, understanding is about knowing the causes. However, in recent years, it has been posed that this is insufficient and there are other factors involved such as a "proposition that specifies the causal relationship that holds between the explanandum and the explanans" [39].

Despite the debates around the limits and objectives of evolution teaching, assessing the understanding of the basic concepts of the TE has been an important topic in science education research. Numerous studies carried out in recent decades have revealed low levels of understanding about the basic principles and mechanisms of biological evolution, not only among the public in general [4, 31, 40] but also among students [1, 13, 31, 41] and teachers [7, 8, 12, 27, 42].

The low levels of understanding in adulthood seems to be related with a deficiency in the evolution teaching in basic educational levels [4]. For this reason, researchers have tried to determine factors that could be hindering teaching and learning of evolution. Some of the barriers that have been found are: 1) teleology [43–45], which is the belief that the occurrences of things in the universe follow a purpose rather than by chance; 2) intentionality [9, 45], which refers to thinking of evolution as a directed process; 3) age [6, 45], for instance, adolescents are more likely to hold Lamarckian views of evolution [45]; and 4) misunderstanding the nature of science [8, 12, 31, 42].

Another significant barrier is religious beliefs. There are studies reporting the existence of a relationship between religious beliefs, levels of religious practice, and acceptance of evolution. [1, 13, 23, 30, 31]. Basically, the students try to apply they preconceived ideas due to their religious background to explain new phenomena because that is what they are familiar with [10]. It has also been found that students who have received a secular education are more likely to understand the theoretical and empirical aspects of the TE as compared with those who received a theological education [15, 16]. Part of these studies highlight the relevance of characterizing religious beliefs and practices, pointing it out as a key aspect to account for the diversity and cultural context of religious traditions. In other words, there is a call to avoid generalizing an alleged conflict between religion and evolution, and to recognize that there is a spectrum of relationships between them. In this way, teachers must be aware of the religious diversity of their students as the starting point to foster scientific knowledge within the classroom [2].

## 1.3 Relationship between acceptance and understanding

Understanding of evolution seems to be related to its acceptance [1, 5, 7, 12–14, 23, 25, 27] but there is still no clear consensus about it.

Some studies report a significant relationship between acceptance and understanding [5, 7, 23, 25]. This relationship is present not only in students but also in teachers. Teachers with a solid base of the basic concepts show a good acceptance and even a preference to teach evolution [5, 7, 8]. Also, Akyol et al. [27] found moderate levels of both acceptance and understanding among college students. This positive correlation suggests that evolution teaching should consider strategies to improve understanding trough acceptance, and the other way around [23].

Paradoxically, research has also reported that good levels of acceptance can be observed together with low levels of understanding of the TE [7, 25]. Some researchers suggest that students can accept the evolution without understanding it and vice versa [14]. In Brazil, Tavares and Bobrowski [25] reported high levels of acceptance of the TE in university students, but very low levels of understanding. Similar results were found by Alters and Nelson [31] where the low levels of comprehension contrast with the good acceptance of university students. The relationship between acceptance and understanding is, at least, complex [5], and some researchers agree that their relationship is nonexistent [1, 12, 13]. The main implication of this would be that not accepting evolution does not prevent to learn it [16], and the TE could be accepted without understand it.

## 1.4 Acceptance and understanding of the TE in Mexico

In Mexico, there is a general idea of the acceptance of the TE among the public, but it is a broad and imprecise panorama. In fact, the UNAM Legal Research Institute carried out a series of surveys in which only 40% of the participants claimed to be convinced that living beings have evolved over time [46]. In 2017, the Mexican National Institute of Statistics and Geography (INEGI) carried out its Survey on the Public Perception of Science and Technology (ENPECYT) with almost 40 million participants, where just over 52% agree that "*Human beings today developed from the evolution of other animal species*" [47]. The number of adults who do not accept evolution is worrying, as it suggests that science teaching is not efficient [6]. In Mexico, the Ministry of Public Education (SEP) establishes two mandatory courses on evolution in basic education: the first of them takes place in the fourth grade of primary school within the subject of *Natural Sciences* and the second one, in the first year of secondary school under the subject of *Sciences I* (biology) (SEP, N. D.). Consequently, high school students have been already exposed to the basic precepts of evolution.

Assessing the current state of acceptance and understanding of the TE in upper secondary education students is of critical relevance: on one hand, high school marks the end of science education for some students and, for others, it lays the groundwork required in college science education [17]. Hence the importance of the present study. Evaluating the acceptance and understanding of the basic concepts of the TE can help in future research in Mexico, providing an overview of biology education, and bringing information that can be comparable with that from other countries. The present paper also contributes to fill the gaps in educational research on evolution in Latin America, through the study of the relationship between understanding and acceptance of the TE in a sample of high school students in the Mexican context.

## 2. Materials and method

### 2.1 Sample

Two questionnaires were implemented in four high school institutions located in the Northeast of Mexico, in the metropolitan area of Monterrey. A total of 370 responses were received from students (178 men and 192 women; ages 15 to 19). The schools where the questionnaires were applied have similar profiles (private schools in areas of medium-high socioeconomic

level). The instruments were applied in English or Spanish, depending on the language used in each school. The questionnaires in English and Spanish were answered by 244 and 160 students, respectively. Importantly, the questionnaires in English are the original versions reported by their authors. The questionnaires in Spanish were generated from three independent translations made by us, considering a natural Mexican phrasing while preserving the original sense of the questions; they can be found in the S1 and S2 Tables in S1 File. A T-test was performed to verify possible significant differences between the results obtained with questionnaires in English and Spanish; no significant differences were found ($\alpha = 0.05$, $p = 0.65$ for I-SEA; $\alpha = 0.05$, $p = 0.14$ for KEE), indicating that the performance of the students was similar irrespectively of the language of the questionnaire. No answers were excluded; all collected data were used for the study.

A demographic section was added to the questionaries to collect information on the student's religious identity; the academic level of their parents, categorized as basic education (primary, secondary and high school), undergraduate, and graduate (master's and doctorate); and, the student's tentative choice of university career, divided into 5 large groups depending on the area of study: Engineering, Biological and Health Sciences, Social Sciences, Arts and Humanities, and Undecided, for those who have not decided yet.

According to the General Health Law on Health Research of Mexico (Ley General de Salud en Materia de Investigación para la Salud) [48], this type of study is classified as "non-risk research" given that no intervention or intentional modification is pursued to the biological, physiological, psychological, or social variables of the individuals. For this reason, an ethical committee is not required. Nevertheless, in our study, we considered the principles and ethical guidelines of the American Educational Research Association [49], complying with the confidentiality of the information and the privacy of the sources, and carrying out a responsible handling of the data. Also, the questionnaires included an explicit statement prior to the questions, in which the student was informed that the data gathered through these instruments was going to be used only for academic purposes and that they could accept or decline their participation in the study just by answering or not the questionnaires.

## 2.2 Instruments

Evaluating the levels of acceptance of the TE has been a central topic of educational research, and during the last decade MATE has become one of the most used instruments for its analysis [50]. Proposed by Rutledge and Warden [21], MATE includes topics on creationism, the age of the Earth, and human evolution, among others. Despite its wide implementation and validation through several statistical analysis, such as principal component analysis, several authors have pointed out that certain questions in MATE combine aspects of acceptance and understanding [18], which can cause inconsistencies when studying the relation between them [51]. Because the aim of this study is to assess acceptance and understanding separately, and to determine if a correlation between them exists, the Inventory of Student Evolution Acceptance (I-SEA) was used in the present study. This 24-question questionnaire, proposed by Nadelson and Southerland [18], evaluates the attitudes of students towards three sub-areas of the TE: macroevolution (questions 1 to 8), microevolution (questions 9 to 16), and human evolution (questions 17 to 24), which allows analyzing the results in more detail at different levels of the TE [24]. Following the Barnes et al. [51] recommendation it is important highlight that using the I-SEA implies to assume its acceptance definition: "the examination of the validity of the knowledge supporting the construct, the plausibility of the construct for explaining phenomenon, persuasiveness of the construct and fruitfulness or productivity of the empirical support for the construct." (p. 1639). That is relevant to discuss comparison across studies. The

**Table 1. Scales of acceptance and understanding of the TE proposed by Kuschmierz et al [38].**

|  | I-SEA | I-SEA Sublevels | KEE |
|---|---|---|---|
| **Very high** | 106–120 | 35–40 | 10 |
| **High** | 91–105 | 30–34 | 8–9 |
| **Moderate** | 76–90 | 25–29 | 6–7 |
| **Low** | 61–75 | 20–24 | 4–5 |
| **Very low** | 24–60 | 8–19 | 0–3 |

questionnaire in Spanish used in the present study can be found in S1 Table in S1 File; the questionnaire in English was the original version reported by Nadelson and Southerland [18].

The questions seek to know the position of the respondents to different statements and can be answered with the options "*Completely agree*, *Agree*, *Neither agree nor disagree*, *Disagree* and *Completely disagree*". The answers are evaluated with integer numbers ranging from 1 (completely disagree) to 5 (completely agree). The I-SEA contains a series of inverted statements (negative sentences), where a high score means skepticism; the score of these statements was inverted to perform the analysis, as it has been done in the literature. Following the recommendations of Betti et al. [24], all questions were scored the same weight. The scores obtained in the questions were added to give a total score per student ranging from 24 to 120 points. A higher global score indicates an overall higher level of acceptance of the TE. Kuschmierz et al. [38] suggest categorizing the level of acceptance into five levels (very low, low, moderate, high, and very high) according to the total score obtained in the questionnaire (see Table 1).

Regarding the understanding of the TE, different instruments have been developed such as the CINS [52], ECKT [21], ACORN [53], and ORI [41], among others. Unfortunately, due to the wide variety of instruments and the fact that the evaluation metrics differ significantly, comparing the results is not always possible [38]. Taking this into account, in the present study the Knowledge of Evolution Exam (KEE) was applied [30] not only because it has been widely used, but also because it has been combined with other instruments to measure acceptance [13, 15], as it is our case. Moreover, the KEE was especially designed for high school students, as it is our case. It consists of a 10-question questionnaire that explores the core concepts of biological evolution such as adaptation, variation, natural selection, evidence of evolution, and macroevolution. It is worth mentioning that this test measures some concepts around evolution. Thus comparisons across researchs, using other test, should consider implicit limitations fo each instrument. Importantly, these, topics are included in the high school Mexican syllabus. The complete questionnaire in Spanish can be found in S2 Table in S1 File; the questionnaire in English was the original version reported by Moore et al. [30]. Each question of KEE has five options to answer and only one is correct, so the score in each item can be only 1 or 0. All the questions were given the same weight, and the total score was the addition of the scores of all questions. Comprehension categories were assigned according to the equivalence table proposed by Kuschmierz et al [38], shown in Table 1.

## 2.3 Data analysis

For both instruments, Cronbach's α (alpha) was used for validation and analysis of their internal consistency. The value of α varies between 0 and 1; the closer it is to 1, the greater the internal consistency of the items [54]. The internal consistency of the I-SEA questionnaire yielded Cronbach's $\alpha = 0.87$, which indicates a high level of reliability [55], for KEE Cronbach's $\alpha = 0.38$, that represent a low level of reliability. Also, to measure the reliability of KEE, the Kuder-

Richardson test was used (KR$_{20}$); it is recommended for multiple-choice tests in which each item is only scored in a binary fashion as "correct" or "incorrect" [55]. Although KEE has been widely used and it is considered a reliable instrument [11, 16, 17, 23, 30, 40], in the present study a low level of internal consistency was found globally for this instrument (KR$_{20}$ = 0.38). We noted that Gefaell et al. [13] reported similar results. We analyzed each item independently and found that several items had good consistency, therefore, it was considered worthwhile to carry out the analysis.

A Pearson statistical test was performed to evaluate the relationship between acceptance and understanding of the TE. It ranges from– 1 to 1. A value of $r$ in the range $-1 < r < 0$ indicates a negative relationship while a value of $r$ in the range $0 < r < 1$ indicates a positive correlation; $r = 0$ indicates that there is no correlation between the variables; the extreme values, $r = \pm 1$, indicate a perfect correlation between the two variables.

To define the appropriate statistical test to evaluate the impact of demographic factors on the acceptance and understanding of evolution, the normality of the data was verified. Fig 1 shows the histograms of the total score obtained in the two questionnaires, (a) I-SEA and (b) KEE, by the population of students. The solid lines correspond to the fit of the distribution to a standard Gaussian function of the form $y = y_0 + ((A/w)/\sqrt{\pi/2})\exp(-2(x - x_c)^2/w^2)$, where $x$ and $y$ are the variables of the histogram, that is, the score value and the number counts, respectively; $y_0$ is a vertical offset (zero in our case); $A$, $w$, and $x_c$ are the amplitude, width, and central value of the Gaussian function, respectively. The standard deviation of the fitted distribution, $\sigma$, is related to the width as $\sigma = w/2$. As can be seen in Fig 1, both distributions show good normality, with a goodness of the fit of $r^2 = 0.9403$ and $r^2 = 0.9860$ for I-SEA and KEE, respectively.

On the other hand, Fig 2 shows the histograms of the total score obtained in the two questionnaires, (a) I-SEA and (b) KEE, after categorizing the total population of students into religious and nonreligious. The solid lines correspond to the fit of the distribution to the above-mentioned Gaussian function. For I-SEA, the distribution of both categories shows good normality, with a goodness of the fit of $r^2 = 0.9416$ and $r^2 = 0.9192$ for religious and nonreligious, respectively. For KEE, the distribution of scores of the religious students shows good normality

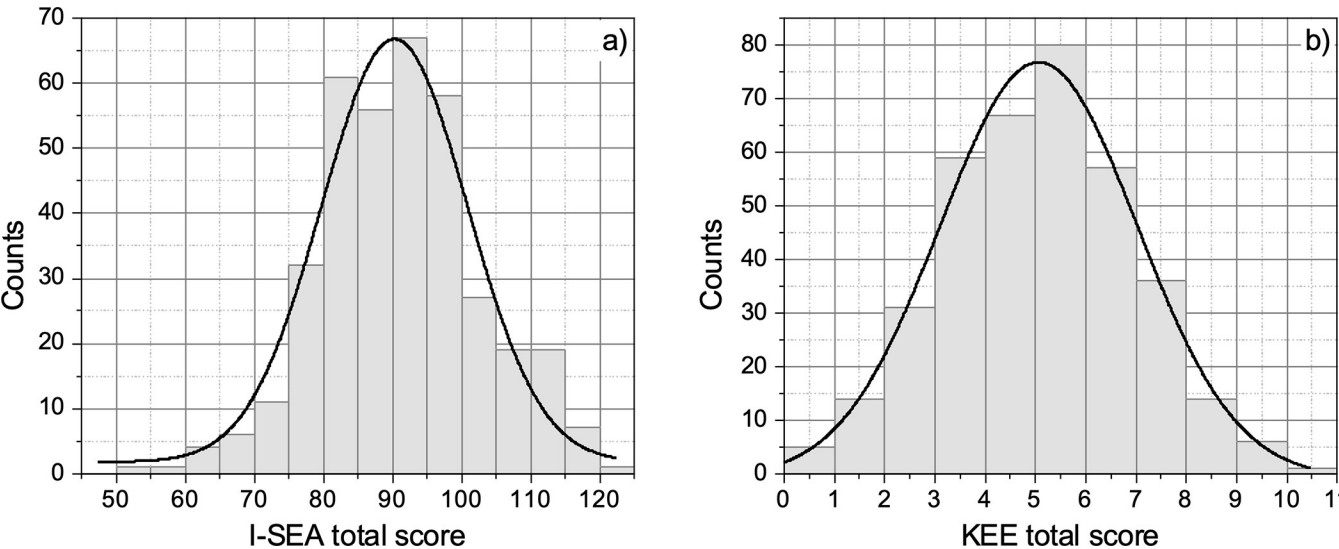

**Fig 1. Histograms of the total score obtained in the two questionnaires, (a) I-SEA and (b) KEE, by the total population of students (N = 370).** The solid lines correspond to the fit of the distribution to a Gaussian function (see text for details).

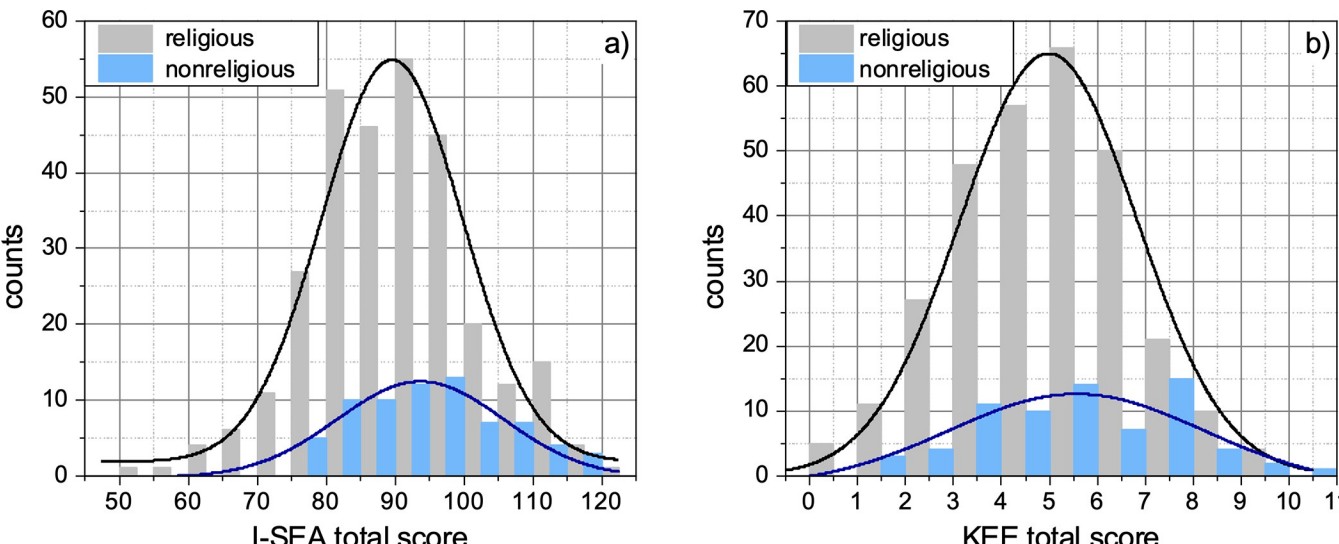

**Fig 2. Histograms of the total score obtained in the two questionnaires, (a) I-SEA and (b) KEE, after categorizing the total population of students into religious (N = 299) and nonreligious (N = 71).** The solid lines correspond to the fit of the distribution to a Gaussian function (see text for details).

($r^2$ = 0.9815), however, for nonreligious students, the distribution shows a well-defined bell shape, but it deviates from an ideal normal distribution ($r^2$ = 0.6834).

In this regard, we note that the variance-based statistical tests that we used are robust against deviations from normality and, as long as the variance of the groups are comparable, the more important requirement is the independence of the groups [56–60]. These conditions are satisfied in our case. There is no dependence between the groups and the standard deviation of I-SEA scores of religious and nonreligious students is $\sigma$ = 11.7519 and $\sigma$ = 10.5247, respectively; as for KEE, the standard deviation of the scores of religious and nonreligious students is $\sigma$ = 1.8226 and $\sigma$ = 2.0615, respectively. Based on this argument, our selection of ANOVA statistical tests is appropriate and valid for data analysis.

Finally, Fig 3 shows the histograms of the total score obtained in the KEE questionnaire, after categorizing the education level of (a) the mother and (b) the father into basic education, college/university, and graduate/postgraduate. The solid lines correspond to the fit of the distribution to the above-mentioned Gaussian function. All distributions show good normality. For the academic level of the students' mother (Fig 3(A)), the fits exhibit goodness of $r^2$ = 0.8855, $r^2$ = 0.9768, and $r^2$ = 0.9620 for the basic education, college, and postgraduate level, respectively. For the academic level of the students' father (Fig 3(B)), the fits exhibit goodness of $r^2$ = 0.8292, $r^2$ = 0.9782, and $r^2$ = 0.9403 for the basic education, college, and postgraduate level, respectively.

In view of the above, to evaluate the potential impact that the demographic factors included in the questionnaire have on the acceptance and understanding of evolution, one-way ANOVA statistical tests were applied to the results of the instruments after classifying the results into the following categories: *Choice of career* (Physical mathematics, Biological sciences and Health, Social Sciences, Arts and Humanities, and Undecided) and *Education of the parents* (Basic Education, University, Postgraduate). This test was used to determine if there were significant differences between the results of the I-SEA and KEE by comparing the different groups with one another within each category (intra-category analysis of significant differences); no inter-category analysis was performed.

Regarding the *Religious identity*, 81% of the participants identified themselves as religious, with Catholicism being the most mentioned (75%); the other religions account for the

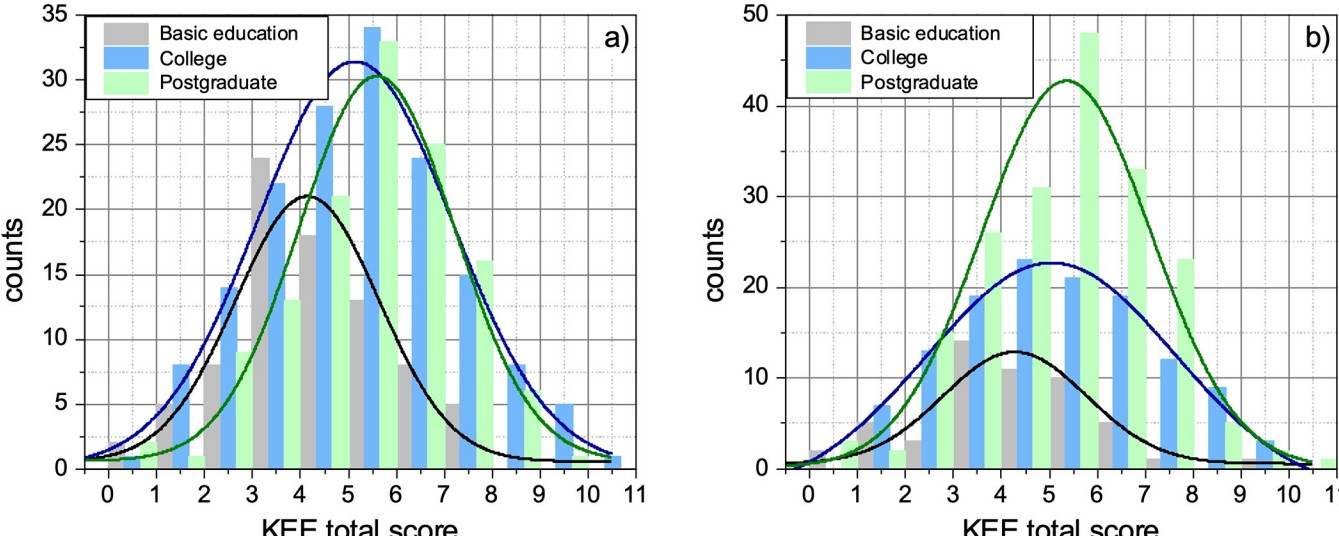

**Fig 3.** Histograms of the total score obtained in the KEE questionnaire, after categorizing the education level of (a) the mother and (b) the father into basic education, college/university, and graduate/postgraduate. The solid lines correspond to the fit of the distribution to a Gaussian function (see text for details).

remaining 6%. The rest of the students (19%) did not manifest a religious belief (atheists and agnostics) (Table 2). Given the disparity in these proportions, it was decided to divide the data into two categories: *Religious* and *Non-religious.* As a result of having only two groups in this case, a two-tailed Student's statistical test was applied to determine the significant differences between them.

## 3. Results

### 3.1 I-SEA results

The overall acceptance of the TE, that is, the average total score of I-SEA questionnaire averaged over all students, was 90.3 (on a scale of 60 to 120 points, SD = 11.6). This level of acceptance is considered moderate according to Kuschmierz et al. [38]. Like other studies in the literature, the level of acceptance was observed to vary for the different sublevels (Fig 4). Recall that each evolution subgroup (microevolution, macroevolution, and human evolution) had 8 questions within the questionnaire, so the maximum score for each one was 40 points (Table 1). To establish the level of acceptance of each category, an analysis of proportions was performed (Table 3). It was found that microevolution is the sublevel with the lowest level of acceptance (29.3) while macroevolution (30.7) and human evolution (30.2) reported slightly

**Table 2. Religious identity distribution.**

| Religious identity | Religion | N (%) |
|---|---|---|
| **Religious** | Catholicism | 276 (75%) |
| | Evangelism | 20 (5.4%) |
| | Judaism | 1 (0.2%) |
| | The Church of Jesus Christ of Latter-day Saints | 1 (0.2%) |
| **No religious** | | |
| | | 71 (19.2%) |

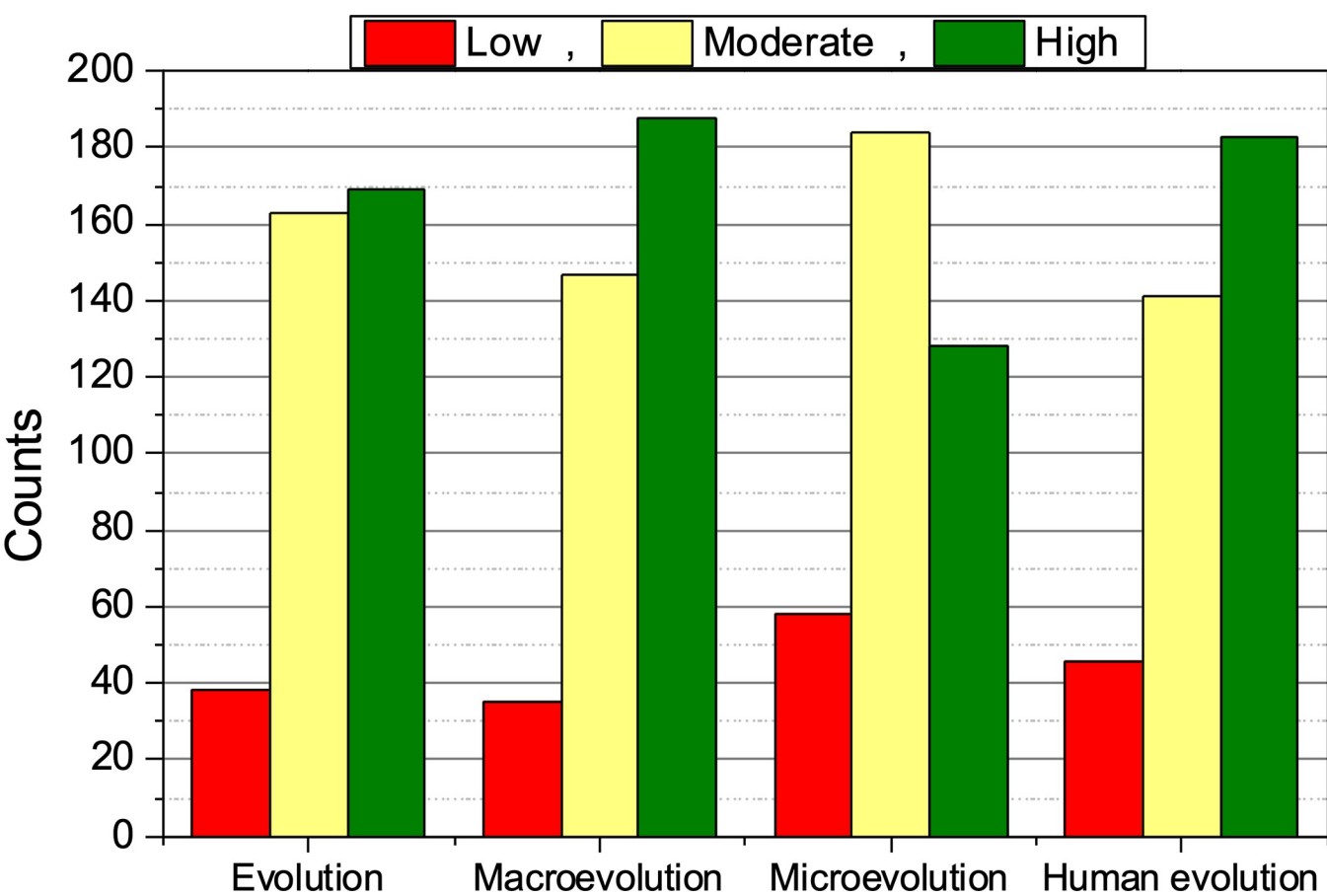

**Fig 4. Evolution acceptance by sublevels.** The values of the subscales were obtained through an analysis of proportions.

higher levels. The summary of the results obtained by the questionnaire by questions are shown in S3 Table and S1 Fig in S1 File.

The levels of acceptance in the different demographic groups did not show significant differences between different groups in the students' career choice ($\alpha = 0.05$, $p = 0.07$), the father's education ($\alpha = 0.05$, $p = 0.17$), nor the mother's education ($\alpha = 0.05$, $p = 0.09$). The only factor found to influence the acceptance of the TE is religious identity ($\alpha = 0.05$, $p = 0.003$). The variables reported, $\alpha$ and $p$, correspond to the significance level and the probability to reject the null hypothesis ($p$-value) in the ANOVA and t-.test. Students who identified themselves as religious exhibited lower acceptance as compared to non-religious students (Table 3). The same result holds when the analysis is done by sublevels: non-religious students have higher scores in all three sublevels; however, significant differences arise only in microevolution ($\alpha = 0.05$, $p = 0.01$) and human evolution ($\alpha = 0.05$, $p = 0.02$).

**Table 3. Evolution acceptance comparison between religious and no religious participants.**

|  | Religious | No religious | Student's t-test ($\alpha = 0.05$) |
|---|---|---|---|
| **Evolution** | 89.6 | 93.2 | 0.003 |
| Macroevolution | 30.6 | 30.9 | 0.69 |
| Microevolution | 29 | 30.3 | 0.01 |
| Human evolution | 29.8 | 32 | 0.02 |

**Table 4. KEE results by question.**

| Item | the TE principles involved | Percentage of correct answers | Most popular answer |
|------|----------------------------|-------------------------------|---------------------|
| 1 | Evidence of evolution | 59% | **E (59%)**[*] |
| 2 | Identification of natural selection in altered environments | 70% | **B (70%)**[*] |
| 3 | Understanding Differential Breeding | 32% | C (39%) |
| 4 | Steps for adaptation | 39% | **C (39%)**[*] |
| 5 | Definition of natural selection | 45% | **D (45%)**[*] |
| 6 | Genetic evidence of a common ancestor | 58% | **B (58%)**[*] |
| 7 | Natural selection as a nonrandom process | 44% | **B (44%)**[*] |
| 8 | Definition of evolution | 45% | **B (45%)**[*] |
| 9 | Mutations as a source of genetic variation | 39% | **B (39%)**[*] |
| 10 | Natural selection as one of the mechanisms that result in evolutionary changes | 26% | B (32%) |

[*] The most popular answer is also the correct one

## 3.2 KEE results

Overall, the data show a low level of understanding of the basic concepts of the TE, with an average of 4.5 out of 10 possible points (Table 4).

The data show a poor understanding of the concepts of natural selection. Although almost 70% of the students were able to recognize natural selection in altered environments (question 2), only 45% of them chose the correct definition for natural selection (question 5). Moreover, the results of question 10 reveal a confusion between evolution and natural selection: only 26% of the students consider natural selection as a mechanism that can result in an evolutionary process and a larger number of students (32%) conceive evolution at the level of individuals and not of populations. This can also be seen in question 3, where almost 39% of the students considered that evolutionary success is based on the individual being able to overcome diseases. Only 32% of students think about it in terms of differential reproduction. Alternative ideas are present at different levels of evolution. Some students fail to identify variations within species and how some characteristics provide an evolutionary advantage over the rest (question 4), and although almost 39% respond in a way correct to the question, there are 35% of students who think of evolution as "intentional" characteristics that the individual can develop depending on their needs. Even so, the majority (about 58%) recognize some evidence that supports the TE (question 1), as well as genetic evidence that suggests a common ancestor (question 6).

The results were also analyzed based on the above-mentioned demographic categories. The ANOVA tests did not show significant differences for the *Choice of career* ($\alpha = 0.05$, $p = 0.15$), but they did for the *Educational level of the parents* (father: $\alpha = 0.05$, $p = 0.001$; mother: $\alpha = 0.05$, $p = 0.00003$). Once again, the variables reported, $\alpha$ and $p$, correspond to the significance level and the *p*-value. Students whose parents have university and graduate level education obtained better scores than those whose parents had only a basic education. Despite the differences between the categories, the understanding of the TE remains at low levels. This can be corroborated in Table 5. When a combined panorama is observed between both parents, the trend is maintained (Table 6), showing that comprehension increases when at least one of the parents has higher education.

Regarding *Religious identity*, the Student's t-test also showed significant differences between religious and non-religious groups ($\alpha = 0.05$, $p = 0.02$). Students who identified as non-religious obtained higher scores (5.1 points) than those who are religious (4.4 points), however, the results reflect a low level of understanding in either case.

**Table 5. Understanding of ET by level of education of a parent.**

|  | *Father* | *Mother* |
|---|---|---|
| **Basic education** | 3.7 | 3.8 |
| **College** | 4.6 | 4.7 |
| **Postgraduate** | 4.8 | 4.9 |

## 3.3 Relationship between I-SEA and KEE results

To establish whether there is a relationship between acceptance and understanding of the TE, a Pearson correlation test was performed. The results are summarized in Fig 5 and show a positive, weak relationship ($y = 2.1119x + 80.656$, $r = 0.34$) between both variables. In Fig 5 a positive trend can be appreciated where acceptance increases as understanding improves, as indicated by the positive slope resulting from the linear regression analysis. However, it is also evident that there exist cases where students have a high level of acceptance and a low level of understanding, and vice versa. Probably other factors influence both variables.

## 4 Discussion

Rice et al. [23] discuss underlying aspects of the understanding and acceptance of the TE. However, there is not a simple conclusion to this discussion and, in fact, it is a complex scenario in which several factors are involved. The data collected in their study suggest that one of the factors related to the acceptance of the TE is religious identity, which is consistent with previous research as well [2, 5, 13, 17, 23, 24, 29–31]. Mexican students in the sample show moderate levels of general acceptance of the TE, which agrees with observations among the public in Mexico [46, 47] and other parts of the world [2, 7, 13, 15, 18, 24, 25]. However, when we filter the results by religious identity, the results show something else. Non-religious students, or those who do not identify with a religion, have higher scores, placing them at a higher level as compared to those who do identify with a religion, who remain at a moderate level.

Some religious beliefs seem to play an important role in the views on evolution [61]. Downie and Barron [28] conducted a 12-year investigation with high school students, and they realized that those who rejected the TE accepted a religious version that would explain the origin of the species. Still, not all religions present the same rejection of the TE [6]. For instance, some Judeo-Christian and Muslim religions express strong resistance to the central ideas of the TE [6, 24, 29]. In Mexico, 80% of the population is Catholic [62], a religion that is also based on the idea of creationism. However, its practitioners do not usually take the Bible narrative about the origin of species literally, but they rather create their own beliefs about it [61], and it seems not to have such a marked conflict with the idea of change in species. Several studies in traditionally Catholic countries show acceptance ranges from moderate to high, such as in Spain [13, 14] and Brazil [25]. Still, a negative influence continues to be reported, as it is also the case in the present study.

Although the conflict perceived by students between their religious beliefs and the TE can negatively affect acceptance [32], they are willing to accept some of the assumptions of the TE

**Table 6. Understanding of TE according to the academic level of the parents.**

|  | *Basic education* | *College* | *Posgraduate* |
|---|---|---|---|
| **Basic education** | 3.5 | - | - |
| **College** | 4.5 | 4.6 | - |
| **Postgraduate** | 4.1 | 4.9 | 4.8 |

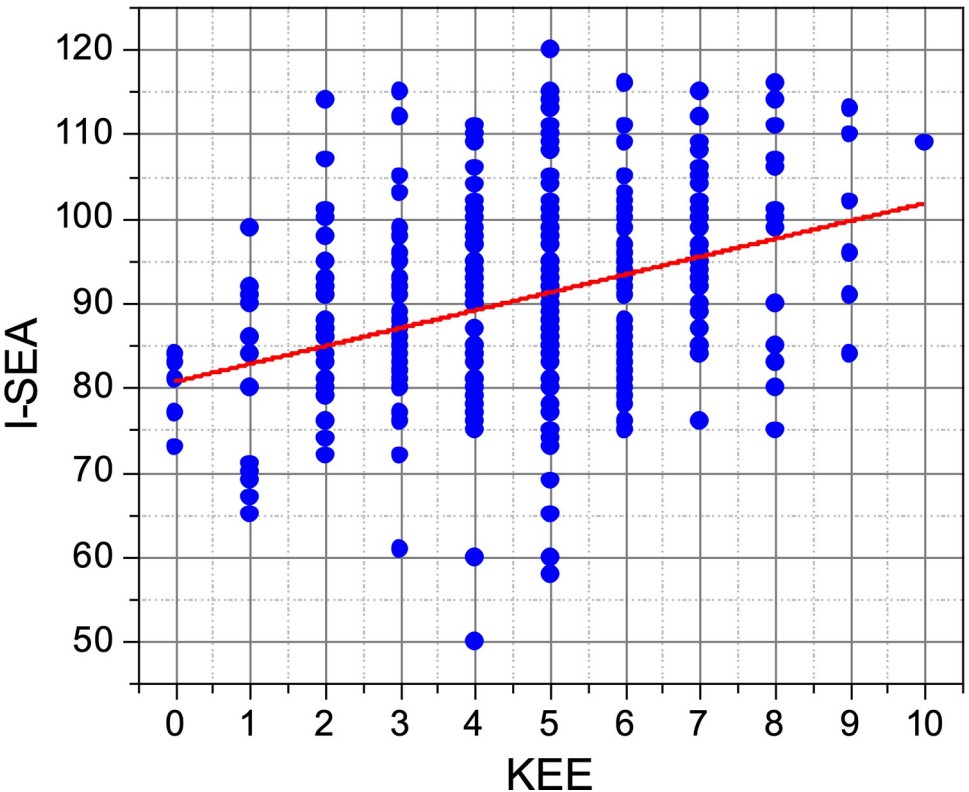

**Fig 5. Relationship between acceptance and understanding.**

[29], although not all in the same way. In the present study it was found that general acceptance of each of the components of the TE ranges from moderate [microevolution) to high [macroevolution and human evolution) and is maintained even when the groups are separated by religious identity. According to the literature, those who reject the TE do not usually have problems accepting that there are changes within the species, but there is an objection to the idea of the origin of new species [28], which means that there is a conflict especially with macroevolution and human evolution. Even though various studies suggest that microevolution is the component of the TE with the best acceptance [18, 24, 28, 29], the results in the present study show the opposite: microevolution had the lowest acceptance scores among the students followed by human evolution and then by macroevolution, with the highest score. By doing a more detailed analysis of the results, we note that this difference might be due to the translation of the questionnaire into Spanish, specifically, because of the use of negative statements pursuing negative answers. Specifically, question 2 ("I think the fossil evidence that scientists use to support the TE is weak and inconclusive"), question 6 ("There is little or no observable evidence to support the theory that describes how a species of organism evolves from a different ancestral form"), question 9 ("I think that organisms, as they exist now, are perfectly adapted to their natural environments and, therefore, will not continue to change"), question 12 ("The species were created to adapt perfectly to their environment, therefore they do not change"), question 13 ("I do not accept the idea that a species of organism will develop new traits over time"), question 15 ("Species exist today in exactly the same appearance and form in which have always existed"), question 18 (Although humans can adapt, humans have not/do not evolve), and question 22 ("Humans do not evolve; they can only change their behavior") are a series of questions written in such a way that the that it was sought that the students

would answer "completely disagree". These questions show the highest peaks of rejection of the TE (see S1 Fig in S1 File), negative sentence structure is more difficult to understand than positive sentence structure [63]. It is also worth pointing out that questions 9, 12, 13 and 15, where this problem was found, are part of the microevolution sublevel, which could be one of the reasons why the acceptance of microevolution differs from that reported in the literature. The way in which the statements are written might influence the result and lower the acceptance levels of the entire sample. Moreover, the students were requested to answer all questions, therefore, it is possible that they have randomly answered some of them, always choosing the answer "Completely agree" or "Agree", without considering the inverted structure of the sentence. An example of this is question 15, which asks about the immutability of species over time, where more than 55% of the students indicated that they agree that species do not change, placing the question with the highest percentage of rejection of the TE; however, in contradiction to this, in question 10, more than 80% of students agree that all groups will continue to change and, in question 11, 81% also agree that there is evidence to suggest that species have changed over time. This inconsistency in the answers suggests that some restructuring is necessary to apply the questionnaire in Spanish. Although the instrument works and the reliability indices are acceptable, negations and inverted sentences can give biased and inconclusive results. The understanding of the TE observed in the present study is low, with results similar to those reported in other investigations [11, 13, 17, 23, 30, 64]. The data show a lack of knowledge of the basic notions of natural selection (questions 3, 5, 7 and 10), variations within the species (questions 4 and 9), and evolution in general (questions 6 and 8), although they can identify evidence that supports the TE (question 1) and the change of organisms when it comes to altered environments (question 2).

In question 1, more than 58% of the students recognize biochemistry, artificial selection, comparative embryology, and vestigial structures as evidence that changes occur in species. In the case of question 2, almost 70% of the students were able to identify a relationship between the alteration of the environments and evolution by natural selection. Similar results were observed by Gefaell et al [13], where they also applied the KEE questionnaire in Spanish (to university students in their case) and obtained the highest percentage of correct answers. Students show higher levels of understanding when it comes to organisms with which they are familiar [10, 65], as well as environmental alterations by mankind that they can relate to. The questions with the lowest percentage of correct answers are question 3, which talks about differential reproduction (32%), and question 10, which focuses on the relationship between natural selection and evolution (26.48%). In both questions, there is evidence of a gap of knowledge about natural selection, what it is and how it works. Also, in the present study it can be appreciated that the students' vision of evolution is based on an individual-centered reasoning, which represents an obstacle for learning [43]. Students think of evolution in terms of individuals evolving (question 10), and that their evolutionary success is based on overcoming obstacles during their lives, such as recovering from illnesses (question 3). These alternative ideas can be attributed to the inability to distinguish between processes experienced by organisms individually and those experienced by populations [65].

The poor understanding of the TE results not only from the difficulties in teaching and learning, but also from informal and deeply ingrained ideas of understanding the world, that usually oppose with scientific explanations [33]. Among these difficulties, religious visions are considered to significantly hinder the learning of the TE because in many cases they constitute sources of poorly formed of concepts that conflict with scientific ideas [66]. In this sense, the results of the present study show a generalized low understanding of the TE, even after analyzing the data separately by religious identity; however, there are significant differences between the scores obtained by religious and non-religious students.

According to Soria [67], religion is the most influential factor in the understanding of the TE, especially when it comes to the origin of mankind. Personal worldviews conflict with scientific content, and, as in this work, other researchers have reported religion as negatively related to evolution [1, 13, 15, 16, 25, 27, 30, 64]. Although some differences were found between the groups by religious identity in the present study, religious and non-religious groups exhibit low understanding. This suggests that the problem per se does not lie in religious belief, but rather in the difficulties of evolutionary reasoning that may be influencing by these beliefs [68]. These inherent cognitive difficulties of the human being are not only affected by religion, but also by cultural factors that shape evolutionary thought patterns [69].

The cultural background of the families plays a transcendental role in the education of students [70], and the understanding of the TE may be related to the educational level of the parents [5]. In the present study, it was found that as the educational level of the parents increased, the students' understanding of the TE also increased, resulting in that the students whose parents have graduate education obtained the highest scores. In this regard, it has been reported that the academic level alone is related to the understanding of the TE: those adults with university degrees and especially with scientific careers have better understanding of the elements of the TE [4], they usually have positive attitudes towards science and are open to questioning their own ideas and listening to other points of view [1]. So, in general, it could be said that the family (and in this case the schooling of the parents) positively influence the understanding of the TE [1, 5, 70].The results of the Pearson correlation analysis show that there is a direct, positive relationship between the acceptance and understanding of the TE ($r = 0.34$). Similar results, indicating that when one improves the other one also does, can be found in several reports in the literature [5, 7, 23, 25, 71]. Although this positive trend can be clearly appreciated in Fig 2, several points fall far from it, where students have good acceptance but low understanding, and vice versa. In general, it is well accepted that the students' points of view are significantly influenced by the knowledge they have on the subject [71]. However, regarding evolution, students can accept the TE without understanding it, or the other way around [14]. These observations have led researchers to conclude that knowledge is a necessary but insufficient condition to improve the acceptance of the TE [34]. In more drastic cases, the relationship between acceptance and understanding is completely rejected [1, 12–14]. The data obtained in the present study suggest that, although a positive, weak relationship between acceptance and understanding was observed, there might be other major factors involved that strongly influence both variables, which complicates the relationship between them.

## 5 Conclusions

The results of the present study show a high level of acceptance and a moderate level of understanding among participants. Students do not accept all components of evolution in the same way and there may be differences between microevolution and macroevolution. Both the understanding and acceptance of TE are influenced by cultural factors, mainly religious identity, resulting in negative effects. Understanding was also observed to be influenced by parental education. Unfortunately, drawing more solid conclusions is hard due to the internal inconsistency of some items in the KEE questionnaire. The correlation between understanding and acceptance of TE is moderate ($r = 0.34$) and suggests that improving one might help to improve the other one. The use of the instruments in this work is still exploratory, it is necessary to adjust the translations of the instruments and expand the samples to validate them and be able to use them with a Hispanic audience.

Based on the results found in the present study, the following general recommendations are made for the teaching of TE: 1) Given the relationship that exists between acceptance and

understanding of the TE, it is necessary to incorporate the improvement of acceptance as part of the structure of the class. 2) It is necessary to highlight that evolution happens in populations, not in individuals. 3) Emphasize the difference between a scientific theory and a religious belief, without facing them as opposite to avoid preconceptions that lead to the rejection of the TE.

## Supporting information

**S1 File.**
(DOCX)

## Author Contributions

**Conceptualization:** Jose Rafael Guzman-Sepulveda, Gonzalo Peñaloza.

**Formal analysis:** Guadalupe Salazar-Enriquez.

**Methodology:** Jose Rafael Guzman-Sepulveda, Gonzalo Peñaloza.

**Supervision:** Jose Rafael Guzman-Sepulveda, Gonzalo Peñaloza.

**Validation:** Jose Rafael Guzman-Sepulveda, Gonzalo Peñaloza.

**Writing – original draft:** Guadalupe Salazar-Enriquez, Jose Rafael Guzman-Sepulveda, Gonzalo Peñaloza.

**Writing – review & editing:** Guadalupe Salazar-Enriquez, Jose Rafael Guzman-Sepulveda, Gonzalo Peñaloza.

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
