## [Decision Letter · Decision Letter 0]

31 Aug 2022

PONE-D-22-19895Understanding and acceptance of the theory of evolution in high school students in MexicoPLOS ONE

Dear Dr. Peñaloza,

Thank you for submitting your manuscript to PLOS ONE. After careful consideration, we feel that it has merit but does not fully meet PLOS ONE’s publication criteria as it currently stands. Therefore, we invite you to submit a revised version of the manuscript that addresses the points raised during the review process.

Please note that we have only been able to secure a single reviewer to assess your manuscript. We are issuing a decision on your manuscript at this point to prevent further delays in the evaluation of your manuscript. Please be aware that the editor who handles your revised manuscript might find it necessary to invite additional reviewers to assess this work once the revised manuscript is submitted. However, we will aim to proceed on the basis of this single review if possible. Their comments are attached below. They request changes to the reliability testing of the questionnaire and the normality evaluation.Could you please revise your manuscript to address their concerns?

We look forward to receiving your revised manuscript.

Kind regards,

Thomas Tischer

Staff Editor

PLOS ONE

Journal Requirements:

"This work was supported by CONACYT (Consejo Nacional de Ciencia y Tecnología) " ext-link-type="uri" xlink:type="simple">https://conacyt.mx/becas_posgrados/"

Reviewers' comments:

Reviewer's Responses to Questions

**Comments to the Author**

1. Is the manuscript technically sound, and do the data support the conclusions?

Reviewer #1: Yes

2. Has the statistical analysis been performed appropriately and rigorously? 

Reviewer #1: Yes

3. Have the authors made all data underlying the findings in their manuscript fully available?

Reviewer #1: Yes

4. Is the manuscript presented in an intelligible fashion and written in standard English?

Reviewer #1: Yes

5. Review Comments to the Author

Reviewer #1: The manuscript was focused on the understanding and acceptance theory of evolution. Authors used validated research tools. Authors used quantitative approach toward obtaining and analyzing of data. The statistical methods were used adequate. The text is divided into chapters typical for empirical study. The text is written in understandable form and also it is written on high level. Below are comments which have got minor character. The comments are presented below.

1. My comments are toward methodological part of the manuscript:

- subchapter “Data analysis” includes kinds of information, which are only informative, for example kinds of information about values of Pearson product moment, what is low and what is high correlation. So please eliminate these kinds of information.

- nowadays, the better statistical methods for the determination of reliability is Cronbach’s alpha coefficient. Even for multiple-choice questions of the test.

- please add the normality evaluation. It confirms the using of selected methods.

2. The results part of the manuscript includes some kinds of information, which regarding to methodology pat like information about reliability.

3. Please revise References according guidelines for authors.

I hope my comments are helpful.

6. PLOS authors have the option to publish the peer review history of their article (what does this mean?). If published, this will include your full peer review and any attached files.

Reviewer #1: **Yes: **Milan Kubiatko

---

## [Author Response · Author response to Decision Letter 0]

21 Sep 2022

Reviewer #1: The manuscript was focused on the understanding and acceptance theory of evolution. Authors used validated research tools. Authors used quantitative approach toward obtaining and analyzing of data. The statistical methods were used adequate. The text is divided into chapters typical for empirical study. The text is written in understandable form and also it is written on high level. Below are comments which have got minor character. The comments are presented below.

We thank the Reviewer for the reading our manuscript and for providing us with pertinent recommendations that allowed us improving the presentation of our work. We have addressed all the comments; all changes can be found highlighted in the revised manuscript.

1. My comments are toward methodological part of the manuscript:

- subchapter “Data analysis” includes kinds of information, which are only informative, for example kinds of information about values of Pearson product moment, what is low and what is high correlation. So please eliminate these kinds of information.

We thank the Reviewer for this observation. We have modified this part of the manuscript and left only the essential information.

- nowadays, the better statistical methods for the determination of reliability is Cronbach’s alpha coefficient. Even for multiple-choice questions of the test.

We thank the Reviewer for this remark. We performed the calculation of the Cronbach’s alpha coefficient for the KEE questionnaire and included the results in the revised manuscript. This reliability value is in good agreement with that obtained using the Kuder-Richardson formalism.

- please add the normality evaluation. It confirms the using of selected methods.

We are particularly thankful to the Reviewer for this comment. Indeed, the verification that the data follows a normal distribution was missing; we apologize for this omission. This information is now included in the revised manuscript. 

Specifically, in the revised manuscript we report the results of numerical fittings to standard Gaussian distributions of: 

i) the scores of I-SEA and KEE for the entire population (Fig. 1 of the revised manuscript).

ii) the scores of I-SEA and KEE after categorizing them based on the students’ religious identity (Fig. 2 of the revised manuscript).

iii) the scores of KEE after categorizing them based on the education level of their mother and father (Fig. 3 of the revised manuscript).

These sets of data are the ones that were used in the statistical tests (t-test and ANOVA), and the distributions show a good normality in general. The only distribution that moves away from an ideal Gaussian distribution is that of the KEE scores of nonreligious students (Fig. 2(b) in the revised manuscript). In this regard, we note that the variance-based statistical tests that we used are robust against deviations from a normal distribution; as long as the variance of the groups are similar, the more important requirement is the independence of the groups [R1-R5], which is satisfied in our case. Based on this argument, our selection of statistical tests is valid. We have clarified this in the revised manuscript and included pertinent references.

[R1] Pearson (1931). The analysis of variance in cases of non-normal variation. Biometrika, 114-133.

[R2] Box (1953). Non-normality and tests on variances. Biometrika, 40(3/4), 318-335.

[R3] Glass, et al., (1972). Consequences of failure to meet assumptions underlying the fixed effects analyses of variance and covariance. Review of educational research, 42(3), 237-288. 

[R4] Harwell, et al., (1992). Summarizing Monte Carlo results in methodological research: The one-and two-factor fixed effects ANOVA cases. Journal of educational statistics, 17(4), 315-339. 

[R5] Lix, et al., (1996). Consequences of assumption violations revisited: A quantitative review of alternatives to the one-way analysis of variance F test. Review of educational research, 66(4), 579-619.

2. The results part of the manuscript includes some kinds of information, which regarding to methodology pat like information about reliability.

We thank the Reviewer for this observation. We have placed that information in the correct section in the revised manuscript.

3. Please revise References according to guidelines for authors.

We thank the Reviewer for this observation. In the revised manuscript, we have amended the format of the references according to the journal’s requirements.

---

## [Decision Letter · Decision Letter 1]

21 Oct 2022

PONE-D-22-19895R1Understanding and acceptance of the theory of evolution in high school students in MexicoPLOS ONE

Dear Dr. Peñaloza,

Thank you for submitting your manuscript to PLOS ONE. After careful consideration, we feel that it has merit but does not fully meet PLOS ONE’s publication criteria as it currently stands. Therefore, we invite you to submit a revised version of the manuscript that addresses the points raised during the review process.

If applicable, we recommend that you deposit your laboratory protocols in protocols.io to enhance the reproducibility of your results. Protocols.io assigns your protocol its own identifier (DOI) so that it can be cited independently in the future. For instructions see: https://journals.plos.org/plosone/s/submission-guidelines#loc-laboratory-protocols. Additionally, PLOS ONE offers an option for publishing peer-reviewed Lab Protocol articles, which describe protocols hosted on protocols.io. Read more information on sharing protocols at https://plos.org/protocols?utm_medium=editorial-emailutm_source=authorlettersutm_campaign=protocols.

We look forward to receiving your revised manuscript.

Kind regards,

Milan Kubiatko

Guest Editor

PLOS ONE

Additional Editor Comments (if provided):

Dear authors

Please revise manuscript according comments of reviewer

Take care

Reviewers' comments:

Reviewer's Responses to Questions

**Comments to the Author**

1. If the authors have adequately addressed your comments raised in a previous round of review and you feel that this manuscript is now acceptable for publication, you may indicate that here to bypass the “Comments to the Author” section, enter your conflict of interest statement in the “Confidential to Editor” section, and submit your "Accept" recommendation.

Reviewer #2: (No Response)

2. Is the manuscript technically sound, and do the data support the conclusions?

Reviewer #2: Partly

3. Has the statistical analysis been performed appropriately and rigorously? 

Reviewer #2: No

4. Have the authors made all data underlying the findings in their manuscript fully available?

Reviewer #2: Yes

5. Is the manuscript presented in an intelligible fashion and written in standard English?

Reviewer #2: Yes

6. Review Comments to the Author

Reviewer #2: Understanding and acceptance of the theory of evolution in high school students in Mexico

The manuscript seems well organized and logically structured traditionally according to IMRAD form. It is written clearly enough to be accessible to non-specialists. Authors have a broad outlook about the field they are interested in, which is declared by 70 references predominantly from the last two decades. The main claims of the paper are relevant and contribute to the dissemination of knowledge about students´ acceptance and understanding of evolutionary theory.

The approach to solving the problem is not original. A similar view on the field of study has e.g. Kuschmierz et al. (2020) searching the students´ acceptance and understanding of evolutionary theory across Europe. This study can bring new information about Mexican students' attitudes and knowledge in this subject matter. In the research, there were used two instruments: the I-SEA questionnaire and the KEE test, but in 2.3 part the MATE was mentioned too, probably, because the scale in the I-SEA questionnaire was adapted to MATE categories according to Kuschmierz et al. (2020, p.6), who remarked that „... the I-SEA, we did not find any suggestions for categories in the original publication, which is why we suggested categories based on the MATE.“. What was the reason to choose these two instruments, if there are so many others specialized in the acceptance and understanding of the evolutionary theory/ the theory of evolution? The authors did not give adequate reasons for it.

Another problem could be observed in using the second instrument (the KEE test). If there was used KEE test proposed by Nadelson and Southerland (2012) as it is cited in the manuscript, then the items/statements differ in the original instrument and instrument used by the authors. They were not verbatim taken over. It seems like pettiness, but...

for example (part 4.1 in manuscript versus Nadelson and Southerland (2012, p. 1655-1656)):

- There is little or no observable evidence to support the theory that describes how a species of organism evolves from a different ancestral form.

- I think that organisms, as they exist now, are perfectly adapted to their natural environments and, therefore, will not continue to change.

- The species were created to adapt perfectly to their environment, therefore they do not change.

- I do not accept the idea that a species of organism will develop new traits over time.

- Species exist today in exactly the same appearance and form in which have always existed.

- Although humans can adapt, humans have not/do not evolve.

in original

- There is little or no observable evidence to support the theory that describes how one species of organism evolves from another ancestral form.

- I think that organisms, as they exist now, are perfectly adapted to their natural environments and so will not continue to change.

- Species were created to be perfectly suited to their environment, so they do not change.

- I don´t accept the idea that a spesies of organism will evolve new traits over time.

- Organisms exist today in exactly the same form in which they always have.

- Although humans may adapt, humans have not/ do not evolve.

The other question is, whether the Spanish translation was correct. It is not elaborated, on how many of the participants solved the I-SEA questionnaire and the KEE test in English and how many in Spanish language. Wasn´t any difference in some parameters of them? Then it is questionable whether the data and analysis support the claims.

The following statement is in part 2.4. (Data analysis) not fortunately chosen: „Extensive psychometric analysis was done to validate the use of these Instruments. For both instruments (the I-SEA and the KEE) Cronbachs´ alpha was used.“. Was the Cronbachs´ alpha used for the validity of the instrument?

Was the testing of data normality meaningful? What kind of data can be obtained processing the I-SEA questionnaire and the KEE test? What kind of variables are tested there? Many researchers argued, that the Likert scale can not generate normally distributed data. What about a multi-item Likert scale? Was appropriate to calculate a Pearson correlation coefficient?

I consider the formal mistake, that several titles are the same in different parts of it (e.g. Acceptance of evolution 1.1., 3.1., 4.2.; Understanding evolution 1.2, 3.2, 4.1; Relationship between acceptance and understanding 1.3, 3.3., 4.3.). In the table 3 the last column is signed „T de student“, in the Conclusions part there is written „...there may be differences between macroevolution and macroevolution“ (?)

Barnes et al (2019) (who were not cited in the manuscript) advised of the fact that „... the choice of the instrument used to measure evolution acceptance can influence the results and conclusions of a study.“ and „Different instruments will measure different aspects of evolution acceptance and may then influence the conclusions that researchers will draw.“ Also, the I-SEA questionnaire and examining its sub-scales are cited in this context. This can bring a problem in the research procedure described in this manuscript, which could not be adequately and credibly reproduced.

Conclusion: The manuscript is unsuitable for publication in its present form, but the study itself disposes of a potential that the authors could re-evaluate and choose relevant data for analysis and then resubmit a revised version.

*cited in a review

Barnes, M.E., Dunlop, H.M., Holt, E.A. et al. Different evolution acceptance instruments lead to different research findings. Evo Edu Outreach 12, 4 (2019). https://doi.org/10.1186/s12052-019-0096-z

Kuschmierz, P., Meneganzin, A., Pinxten, R. et al. Towards common ground in measuring acceptance of evolution and knowledge about evolution across Europe: a systematic review of the state of research. Evo Edu Outreach 13, 18 (2020). https://doi.org/10.1186/s12052-020-00132-w

Nadelson, L. S., Southerland, S. (2012). A More Fine-Grained Measure of Students’ Acceptance of Evolution: Development of the Inventory of Student Evolution Acceptance—I-SEA. International Journal of Science Education, 34(11), 1637–1666. doi:10.1080/09500693.2012.702235.

7. PLOS authors have the option to publish the peer review history of their article (what does this mean?). If published, this will include your full peer review and any attached files.

Reviewer #2: No

---

## [Author Response · Author response to Decision Letter 1]

10 Nov 2022

Reviewer #2: 

The manuscript seems well organized and logically structured traditionally according to IMRAD form. It is written clearly enough to be accessible to non-specialists. Authors have a broad outlook about the field they are interested in, which is declared by 70 references predominantly from the last two decades. The main claims of the paper are relevant and contribute to the dissemination of knowledge about students’ acceptance and understanding of evolutionary theory.

We thank the Reviewer for reading our manuscript and providing us with comments that allowed us improving the presentation of our work. We have addressed all the comments; all changes can be found highlighted in the revised manuscript.

The approach to solving the problem is not original. A similar view on the field of study has e.g. Kuschmierz et al. (2020) searching the students´ acceptance and understanding of evolutionary theory across Europe. This study can bring new information about Mexican students’ attitudes and knowledge in this subject matter. In the research, there were used two instruments: the I-SEA questionnaire and the KEE test, but in 2.3 part the MATE was mentioned too, probably, because the scale in the I-SEA questionnaire was adapted to MATE categories according to Kuschmierz et al. (2020, p.6), who remarked that “... the I-SEA, we did not find any suggestions for categories in the original publication, which is why we suggested categories based on the MATE”. What was the reason to choose these two instruments, if there are so many others specialized in the acceptance and understanding of the evolutionary theory/ the theory of evolution? The authors did not give adequate reasons for it.

As explained in our manuscript, one of our goals was to measure the acceptance and knowledge of the TE as independently as possible. The combination of questionnaires used serve this purpose and allows for different types of analysis. For instance, one can identify factors that influence the acceptance and knowledge separately; in our case, we found that religious identity was the only factor, among those considered, that affected the acceptance while the knowledge was affected by both religious identity and the parents’ education. Also, one can reliably stablish the existence of a relation between acceptance and knowledge. In this regard, some questions in MATE assess aspects of acceptance and understanding simultaneously, which can cause inconsistencies when studying the relation between them e.g., generation of stronger correlations. Finally, I-SEA evaluates different sublevels of the TE (microevolution, macroevolution, and human evolution). This was important for us because 1) for the Mexican context, there is not much information about the acceptance of the TE e.g., what aspects are more accepted/rejected, and 2) it allows using the information collected for pedagogic purposes e.g., to implement specially designed classroom interventions. We have clarified this in the discussion of the revised manuscript.

Another problem could be observed in using the second instrument (the KEE test). If there was used KEE test proposed by Nadelson and Southerland (2012) as it is cited in the manuscript, then the items/statements differ in the original instrument and instrument used by the authors. They were not verbatim taken over. It seems like pettiness, but...

for example (part 4.1 in manuscript versus Nadelson and Southerland (2012, p. 1655-1656)):

• There is little or no observable evidence to support the theory that describes how a species of organism evolves from a different ancestral form.

• I think that organisms, as they exist now, are perfectly adapted to their natural environments and, therefore, will not continue to change.

• The species were created to adapt perfectly to their environment, therefore they do not change.

• I do not accept the idea that a species of organism will develop new traits over time.

• Species exist today in exactly the same appearance and form in which have always existed.

• Although humans can adapt, humans have not/do not evolve.

in original

• There is little or no observable evidence to support the theory that describes how one species of organism evolves from another ancestral form.

• I think that organisms, as they exist now, are perfectly adapted to their natural environments and so will not continue to change.

• Species were created to be perfectly suited to their environment, so they do not change.

• I don´t accept the idea that a species of organism will evolve new traits over time.

• Organisms exist today in exactly the same form in which they always have.

• Although humans may adapt, humans have not/do not evolve.

We thank the Reviewer for this comment. The questionnaires in English we used were the original versions reported by their respective authors (KEE: Moore et al., 2009; I-SEA: Nadelson and Southerland, 2012). The comparison made above by the Reviewer is based on questions that were neither written nor reported by us (we suspect that the Reviewer translated our Spanish questionnaire back to English and assumed that we used that new English version). The questionnaires in Spanish were generated from three independent translations made by us, considering a natural Mexican phrasing while preserving the original sense of the questions. We have clarified this in the revised manuscript.

The other question is, whether the Spanish translation was correct. It is not elaborated, on how many of the participants solved the I-SEA questionnaire and the KEE test in English and how many in Spanish language. Wasn´t any difference in some parameters of them? Then it is questionable whether the data and analysis support the claims.

We thank the Reviewer for this comment and apologize for our omission. The questionnaires in English and Spanish were answered by 244 and 160 students, respectively. T-tests were performed on the results of the two groups (English and Spanish), and no significant differences were found (α=0.05, p=0.65 for I-SEA; α=0.05, p=0.14 for KEE), indicating that the performance of the students was similar irrespectively of the language of the questionnaire. We have included this information in the revised manuscript.

The following statement is in part 2.4. (Data analysis) not fortunately chosen: “Extensive psychometric analysis was done to validate the use of these Instruments. For both instruments (the I-SEA and the KEE) Cronbachs’ alpha was used”. Was the Cronbachs’ alpha used for the validity of the instrument?

Yes, both instruments were validated using Cronbach’s α (alpha) as a metric of reliability and internal consistency. For KEE, additionally, the Kuder-Richardson test was performed as it is recommended for multiple-choice tests where each item is scored binarily (correct/incorrect). We have clarified this in the revised manuscript.

Was the testing of data normality meaningful? What kind of data can be obtained processing the I-SEA questionnaire and the KEE test? What kind of variables are tested there? Many researchers argued, that the Likert scale can not generate normally distributed data. What about a multi-item Likert scale? Was appropriate to calculate a Pearson correlation coefficient?

We thank the Reviewer for this comment. We would like to note that the justification of our statistical analysis was amply discussed in the first review. In fact, one major modification in our revised manuscript, was the addition of all details about the normality tests, at the request of the first Reviewer. 

Testing the normality of the data constitutes the validation of our choice of analysis. As the Reviewer correctly points out, there can be situations where the data are not normally distributed; in such cases, the tests carried out would be compromised (especially ANOVA and T-test), and a different analysis would be required.

I consider the formal mistake, that several titles are the same in different parts of it (e.g. Acceptance of evolution 1.1., 3.1., 4.2.; Understanding evolution 1.2, 3.2, 4.1; Relationship between acceptance and understanding 1.3, 3.3., 4.3.). In the table 3 the last column is signed “T de student”, in the Conclusions part there is written “...there may be differences between macroevolution and macroevolution” (?)

We thank the Reviewer for this comment. We have corrected this in the revised manuscript.

Barnes et al (2019) (who were not cited in the manuscript) advised of the fact that “... the choice of the instrument used to measure evolution acceptance can influence the results and conclusions of a study.” and “Different instruments will measure different aspects of evolution acceptance and may then influence the conclusions that researchers will draw.” Also, the I-SEA questionnaire and examining its sub-scales are cited in this context. This can bring a problem in the research procedure described in this manuscript, which could not be adequately and credibly reproduced.

Thank you for referring us to the work of Barnes et al (2019); we found that reference very useful to improve out manuscript. As pointed out by Barnes et al, by using a certain instrument, one implicitly assumes the definition of the constructs used for its development. I-SEA is not the exception as it is based on a particular definition of acceptance. In the revised manuscript, we have explicitly stated such definition and advised the reader to consider the differences in the definition of acceptance when comparing the results obtained by using different instruments. In this sense, it is important to mention that Barnes et al. affirm that no test is more effective than others; they simply advise researchers to be aware of the constructs assumed by each test and its limitations, as we have done in our revised manuscript. Also, Barnes et al. (2019) state that only a few research studies have been carried out using I-SEA. In this sense, we consider that our paper contributes to exploring its use in contexts and populations where it has not been utilized. Moreover, Barnes et al. (2019) pointed out that a positive relation between understanding and acceptance is often encountered when MATE is used. This might be due to some questions in MATE assess aspects of acceptance and understanding simultaneously, which can create an artificial correlation because the instruments overlap and evaluate the same aspect. In our case, we found a positive relationship between understanding and acceptance after measuring them independently, which can be considered a reliable result. 

We have clarified the above in the revised manuscript, including our criteria to select the tests and an explanation why I-SEA was adequate for us (it allows disclosing the level of acceptance/rejection in the Mexican context and designing ad hoc classroom interventions; see comment above).

Conclusion: The manuscript is unsuitable for publication in its present form, but the study itself disposes of a potential that the authors could re-evaluate and choose relevant data for analysis and then resubmit a revised version.

Once again, we thank the Reviewer for reading our manuscript and providing us with feedback that allowed us improving the presentation of our work.

---

## [Editor Report · Decision Letter 2]

21 Nov 2022

Understanding and acceptance of the theory of evolution in high school students in Mexico

PONE-D-22-19895R2

Dear Dr. Peñaloza,

We’re pleased to inform you that your manuscript has been judged scientifically suitable for publication and will be formally accepted for publication once it meets all outstanding technical requirements.

Kind regards,

Milan Kubiatko

Guest Editor

PLOS ONE

Additional Editor Comments (optional):

Good work
---

## [Editor Report · Acceptance letter]

29 Nov 2022

PONE-D-22-19895R2 

Understanding and acceptance of the theory of evolution in high school students in Mexico 

Dear Dr. Peñaloza:

I'm pleased to inform you that your manuscript has been deemed suitable for publication in PLOS ONE. Congratulations! Your manuscript is now with our production department. 

Kind regards, 

on behalf of

Dr. Milan Kubiatko 

Guest Editor

PLOS ONE